# Curvature-Aware Residual Prediction for Stable and Faithful Diffusion Transformer Acceleration Under Large Sampling Intervals

## Abstract

Diffusion Transformers have achieved remarkable performance in generative tasks, yet their large model size and multi-step sampling requirement lead to prohibitively expensive inference. Conventional caching methods reuse features across timesteps to reduce computation, but introduce approximation errors that accumulate during denoising—a problem exacerbated under large sampling intervals where significant feature variations amplify errors. Recent prediction-based approaches (e.g., TaylorSeers) improve efficiency but remain limited by sensitivity to feature variations across distant timesteps and the inherent truncation errors of Taylor expansions. To address these issues, we propose a novel **C**urvature-**A**ware **R**esidual **P**rediction (CARP) framework, which shifts the prediction target from raw features to residual updates within Diffusion Transformer blocks. We observe that residuals exhibit more stable and predictable dynamics over time compared to raw features, making them better suited for extrapolation. Our approach employs a rational function-based predictor, whose theoretical superiority over polynomial approximations is rigorously established: the numerator performs linear extrapolation using adjacent features, while the denominator incorporates discrete curvature to adaptively modulate the strength and behavior of the prediction. This design effectively captures the alternation between gradual refinements and abrupt transitions in diffusion denoising trajectories. Additionally, we introduce a curvature-guided gating mechanism that regulates the use of predicted values, enhancing robustness under large sampling steps. Extensive experiments on FLUX, DiT-XL/2, and Wan2.1 demonstrate our method's effectiveness. For instance, at 20 denoising steps, we achieve up to 2.88× speedup on FLUX, 1.46× on DiT-XL/2, and 1.72× on Wan2.1, while maintaining high quality across FID, CLIP, Aesthetic, and VBench metrics, significantly outperforming existing feature caching methods. In user studies on FLUX, CARP receives nearly 25% more preference than the second-best method. These results underscore the advantages of residual-targeted prediction combined with a rational function-based extrapolator for efficient, training-free acceleration of diffusion models.

## 1 Introduction

Diffusion models( Song & Ermon (2020)) have emerged as the predominant framework for high-fidelity visual generation. The recent shift from convolutional U-Nets( Ronneberger et al. (2015)) to more scalable and expressive Diffusion Transformers (DiTs)( Peebles & Xie (2023)) has markedly enhanced model capacity and representational power, albeit at the expense of a substantial increase in parameter count. This scaling trend results in prohibitively high inference costs, positioning the multi-step denoising process as a critical bottleneck for real-world deployment under strict constraints on latency, throughput, and energy efficiency.

Various techniques have been proposed to accelerate diffusion inference, including quantization( Li et al. (2023a)), pruning( Fang et al. (2023)), and knowledge distillation. Among these, caching-based methods have gained prominence due to their advantage of being training-free and architecture-

preserving. Existing caching strategies for diffusion models fall into two primary categories: (i) **Reuse-based methods:** These approaches improve upon the effectiveness of naive feature reuse by refining the reuse decision or granularity. For instance, TeaCache( Liu et al. (2024)) incorporates input differences and timestep embeddings to better predict output changes and decide when reuse is safe, while ToCa( Zou et al. (2024)) decomposes features at the token level and selectively reuses only the most informative components of the activation tensor. (ii) **Prediction-based methods:** Instead of reusing cached features, TaylorSeer( Liu et al. (2025b)) directly predicts features for future steps via Taylor expansion—a polynomial-based extrapolation method—thereby replacing reuse with an explicit approximation of the next-step representation.

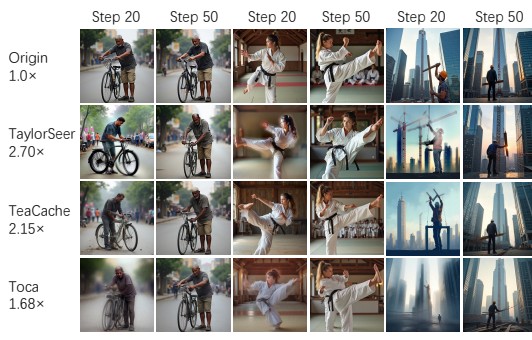

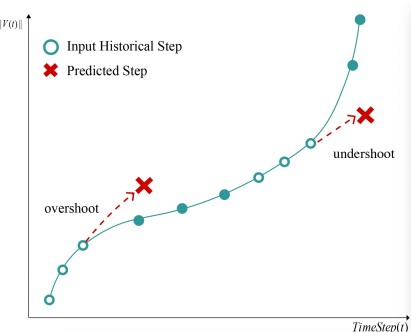

(a) Visualization of different caching mechanisms

(b) Visualization of Polynomial Extrapolation Forecasts

Figure 1: Visualization of different caching mechanisms and polynomial extrapolation forecasts. (a) Comparison of different caching strategies, showing their performance under low denoising steps. (b) Visualization of polynomial extrapolation forecasts, illustrating the error accumulation and deviation from the true trajectory over time.

Despite their promise, these caching methods are typically evaluated under around 50 denoising steps, and their efficacy diminishes significantly in low-step regimes—precisely those most relevant for real-world applications. As illustrated in Figure 1a, when employing a reduced number of denoising steps (i.e., large sampling steps), current approaches suffer significant performance degradation. This decline stems from two key issues: first, low denoising steps lead to an increased time span of the prediction window across adjacent time steps, significantly reducing the feature similarity between adjacent steps, which renders feature reuse-based strategies ineffective. Second, for polynomial prediction methods, the increased feature disparity introduces more volatile dynamic trends, which exacerbates the difficulty of polynomial fitting. Due to the inherent limitations of polynomial approximation, this results in significant deviations from the true trajectory as shown in Figure1b, and severe image distortion during extrapolation.

To address these challenges, we propose a Curvature-Aware Residual Prediction (CARP) framework that estimates evolutionary trends at future steps by leveraging feature residuals from a short historical window through rational function-based extrapolation. The rational function-based predictor consists of a numerator and a denominator: the numerator primarily employs adjacent time-step features to perform a linear extrapolation that serves as the basis of the prediction, while the denominator incorporates the discrete curvature among features within the time window to nonlinearly modulate the strength and behavior of this extrapolation. This formulation enables the model to capture the characteristic alternation between gradual refinements and abrupt shifts that naturally arises in diffusion processes. Additionally, we introduce a curvature-aware weighting mechanism that adaptively allocates dependency between proximal and distal features. This mechanism adjusts the sign and magnitude of the denominator's weight based on the trajectory's curvature, ensuring stable and robust extrapolation across diverse dynamic conditions.

To further mitigate error accumulation in long-horizon forecasting, we propose a shift in prediction target: rather than directly extrapolating high-dimensional feature maps, we predict the end-to-end residuals(defined as the output of the Transformer stack minus its input) within the Diffusion Transformer stack. The residuals capture the "net update" applied by the network at each iteration step, thereby exhibiting a more tractable and predictable structure—a claim supported by both theoret-

ical and empirical evidence. This refinement considerably improves prediction stability. Finally, recognizing that even rational functions are limited in forecasting abrupt regime shifts, we employ trajectory curvature as a dynamic gating signal. This curvature-based trigger provides a principled criterion for adaptively regulating prediction, ensuring stable acceleration across diverse denoising regimes.

Extensive experiments on text-to-image, class-to-image generation, and text-to-video demonstrate the effectiveness of CARP on FLUX, DiT-XL/2, and Wan2.1, over previous feature caching methods. For instance, at 20 denoising steps, we achieve up to 2.88× speedup on FLUX, 1.46× on DiT-XL/2, and 1.72× on Wan2.1, while maintaining high quality with only a 3% loss in FID on FLUX and a 5% loss in VBench2 score, significantly outperforming existing feature caching approaches. In the FLUX user study, CARP receives nearly 25% more preference than the second-best approach. The main contributions of our work are presented as follows:

1. **Curvature-Aware Rational Prediction.** We design a rational-function-based predictor that adapts to the local curvature of residual trajectories, achieving more robust forecasting than polynomial extrapolation in low-denoising-step regimes. It also maintains strong performance with larger denoising steps.

2. **Residual-Targeted Predictive Caching.** We reveal the limitation of feature-based caching under low denoising steps and propose to use *end-to-end residuals* as a more predictable target, supported by both theoretical analysis and empirical evidence.

3. **Comprehensive Validation and Compatibility.** We validate our approach across multiple benchmarks, demonstrating that it significantly improves cache stability and reduces inference cost, all while being entirely training-free and compatible with existing Diffusion Transformers.

## 2 RELATED WORK

This section, we review previous works related to diffusion model acceleration and feature caching techniques. Please refer to Appendix A.3 for details.

## 3 PRELIMINARIES

### 3.1 DIFFUSION MODELS

Diffusion models generate data by reversing a gradual noising process. In the forward process, a clean data sample $\mathbf{x}_0 \sim p_{\text{data}}$ is progressively perturbed with Gaussian noise, producing a sequence $\mathbf{x}_t$ that converges to nearly isotropic Gaussian as $t \to T$. The generative process corresponds to reversing this evolution, i.e., gradually transporting noise back into data. In continuous time, we adopt the *probability–flow ODE* formulation of the reverse dynamics:

$$\frac{d\mathbf{x}_t}{dt} = -\mathbf{v}_\theta(\mathbf{x}_t, t), \tag{1}$$

where $\mathbf{v}_\theta$ is a learned velocity field (deterministic drift). Inference amounts to numerically integrating Eq. 1 from $t{=}T$ to $t{=}0$ starting from Gaussian noise to obtain $\mathbf{x}_0$.

### 3.2 NAIVE CACHE

A simple acceleration heuristic is to *directly reuse* the previous-step feature instead of recomputing it at step $t$:

$$\tilde{\mathbf{h}}_t := \mathbf{h}_{t-1}. \tag{2}$$

A fixed-stride variant reuses an older feature, $\tilde{\mathbf{h}}_t := \mathbf{h}_{t-K}$ with $K \geq 1$. While this saves a forward pass, the assumption of near-invariance across steps breaks under large strides, causing drift and quality degradation.

## 4 CURVATURE-AWARE RESIDUAL PREDICTION(CARP)

### 4.1 OVERALL FRAMEWORK

In this section, we introduce *CARP*, a novel residual-based predictive caching method for accelerating Diffusion Transformer inference. *CARP* offers stable acceleration under low denoising steps,

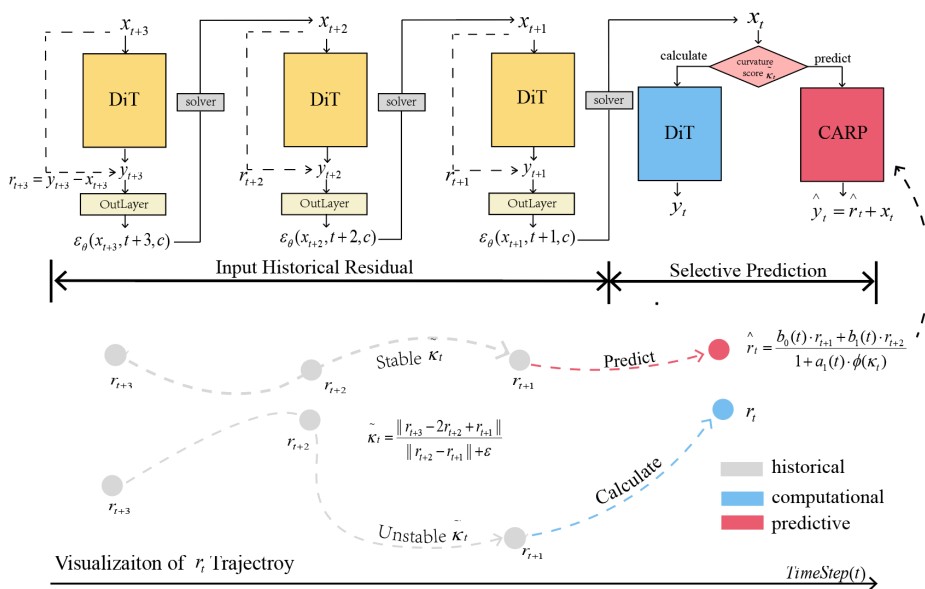

Figure 2: An overview of CARP. It maintains a residual history window of size 3 and caches end-to-end residuals. At time step t, CARP computes the curvature based on the residual history, using its magnitude as a gating mechanism for prediction. When the residual trajectory exhibits complex dynamics, the original computation process is preserved. Otherwise, a rational function is employed to incorporate the curvature information into the linear extrapolation, preventing over/undershooting and achieving more accurate trajectory predictions.

remaining training-free and easily applicable to existing models. We show that using a rational function-based predictor reduces error accumulatio compared to polynomial extrapolation. By leveraging end-to-end residuals, *CARP* captures smoother trajectories, minimizing error accumulation. This approach adapts to residual trajectory behavior, ensuring both accuracy and stability, even under large sampling strides.

## 4.2 CURVATURE-AWARED RESIDUAL PREDICTION

**The Curvature-Aware Predictor in Rational Form.** To robustly handle the complex dynamics of residual trajectories, we introduce a *sign-aware* rational predictor. Given a short residual history $\{\mathbf{r}_{t+3}, \mathbf{r}_{t+2}, \mathbf{r}_{t+1}\}$, we predict the next residual $\hat{\mathbf{r}}_t$ using the following elementwise formulation:

$$\hat{\mathbf{r}}_t = \frac{b_0(t)\mathbf{r}_{t+1} + b_1(t)\mathbf{r}_{t+2}}{1 - a_1(t)\,\tanh(\gamma \cdot \kappa_t)}, \tag{3}$$

where $\kappa_t = \mathbf{r}_{t+3} - 2\mathbf{r}_{t+2} + \mathbf{r}_{t+1}$ is the *elementwise discrete curvature*. The key innovation lies in the denominator, which acts as a bidirectional controller. As we will justify in Section 4.3, the sign of $\kappa_t$ indicates the likely direction of the extrapolation error (overshoot vs. undershoot). By using the $\tanh$ function, our denominator can become greater or less than 1, allowing it to intelligently *damp* predicted overshoots and *boost* predicted undershoots. The coefficients $b_0(t)$, $b_1(t)$, and $a_1(t)$ adapt dynamically to control the prediction strategy and correction intensity.

**Adaptive Coefficient Modulation via Normalized Curvature.** The intensity of our adaptive control is governed by a single, intuitive signal: the *normalized curvature measure* $\tilde{\kappa}_t$. This scalar value quantifies the overall magnitude of non-linearity in the recent trajectory:

$$\tilde{\kappa}_t = \frac{\|\mathbf{r}_{t+3} - 2\mathbf{r}_{t+2} + \mathbf{r}_{t+1}\|}{\|\mathbf{r}_{t+2} - \mathbf{r}_{t+1}\| + \varepsilon}.$$

Based on $\tilde{\kappa}_t$, we modulate the predictor's coefficients at two levels:

**1. Numerator Modulation (Prediction Strategy):** The coefficients $b_0(t)$ and $b_1(t)$ adapt the high-level prediction strategy. When the trajectory is smooth ($\tilde{\kappa}_t \to 0$), we default to an aggressive linear extrapolation (($b_0, b_1$) $\to$ (2, -1)). When it is highly curved ($\tilde{\kappa}_t$ is large), we pivot to a more conservative first-order hold (($b_0, b_1$) $\to$ (1, 0)). This transition is governed by a blending factor $s = \min(\tilde{\kappa}_t/T_b, 1)$:

$$b_0(t) = (1 - s) \cdot 2 + s \cdot 1 = 2 - s$$
$$b_1(t) = (1 - s) \cdot (-1) + s \cdot 0 = s - 1$$

**2. Denominator Modulation (Correction Intensity):** The coefficient $a_1(t)$ controls the *intensity* of the bidirectional correction. The correction should be minimal for smooth trajectories but strong for volatile ones. We thus employ a thresholded linear mapping:

$$a_1(t) = c \cdot \max(0, \tilde{\kappa}_t - T_a), \tag{4}$$

where $T_a$ is a threshold and $c$ is a scaling factor. This ensures that the powerful bidirectional control only activates when the trajectory's volatility warrants it.

**Denoising-Trajectory Adaptive Gating.** While our predictor is highly adaptive, we employ a final safety net for exceptionally chaotic trajectory segments. A high-level gating mechanism, controlled by the same normalized curvature $\tilde{\kappa}_t$, decides whether to predict or recompute. Given a gating threshold hyperparameter $N$:

- **Predict (low to moderate curvature):** If $\tilde{\kappa}_t < N$, the trajectory is deemed controllable. We use our sign-aware rational predictor in Eq. equation 3 to forecast $\hat{\mathbf{r}}_t$.

- **Recompute (high curvature):** If $\tilde{\kappa}_t \geq N$, the trajectory is too unstable even for our adaptive controller. We perform a full forward pass to guarantee accuracy and prevent error propagation.

This multi-layered strategy—adapting the prediction method, applying bidirectional correction, and using a hard gate for stability—allows CARP to robustly balance acceleration and fidelity. For implementation details, see Appendix A.2.

### 4.3 ERROR ANALYSIS OF THE CURVATURE-AWARE RESIDUAL PREDICTOR

In this section, we first establish the theoretical link between curvature sign and error direction, then use this insight to design a *curvature-aware* residual predictor.

**The Link Between Curvature Sign and Error Direction.** The key to bidirectional control lies in diagnosing the direction of the polynomial prediction error. Let $\hat{r}_0^{\text{poly}}$ be the baseline linear extrapolator. Its local truncation error, $E_{\text{poly}} = \hat{r}_0^{\text{poly}} - r_0$, is driven by the trajectory's derivatives. From Taylor's theorem, we have:

$$E_{\text{poly}} \approx -\frac{\Delta^2}{2} r''(t). \tag{5}$$

Simultaneously, our elementwise discrete curvature $\kappa_t = r(t + 3\Delta) - 2r(t + 2\Delta) + r(t + \Delta)$ is a finite-difference approximation of the second derivative:

$$\kappa_t \approx \Delta^2 r''(t). \tag{6}$$

Combining these two approximations reveals a crucial insight:

$$\mathbb{E}[E_{\text{poly}}] \propto -\kappa_t \quad \implies \quad \text{sign}(E_{\text{poly}}) \approx -\text{sign}(\kappa_t). \tag{7}$$

This relationship provides the diagnostic tool we need. The sign of the computable curvature $\kappa_t$ reliably indicates the sign of the prediction error.

- If $\kappa_t > 0$ (trajectory is concave up), the error $E_{\text{poly}}$ is likely negative, meaning $\hat{r}_0^{\text{poly}} < r_0$. This is a predictive **undershoot**. We need to *boost* the prediction.
- If $\kappa_t < 0$ (trajectory is concave down), the error $E_{\text{poly}}$ is likely positive, meaning $\hat{r}_0^{\text{poly}} > r_0$. This is a predictive **overshoot**. We need to *damp* the prediction.

**The Curvature-Aware Rational Controller.** Armed with this insight, we redesign the rational predictor's denominator to be sign-aware. Instead of using the absolute value of curvature, we use its sign to determine the direction of control. Our proposed predictor remains $\hat{r}_t = \hat{r}_0^{\text{poly}}/D_t$, but the denominator $D_t$ is now formulated to be greater or less than 1:

$$D_t = 1 - a_1(t) \cdot \tanh(\gamma \cdot \kappa_t), \tag{8}$$

where $\gamma$ is a scaling hyperparameter that controls the sensitivity to curvature, and $\tanh(\cdot)$ is the hyperbolic tangent function. This formulation has several key advantages:

1. **Bidirectional Control:** When $\kappa_t > 0$ (undershoot), $\tanh(\cdot)$ is positive, making $D_t < 1$. Dividing by a number less than 1 *boosts* the magnitude of $\hat{r}_0^{\text{poly}}$, correcting the undershoot. Conversely, when $\kappa_t < 0$ (overshoot), $\tanh(\cdot)$ is negative, making $D_t > 1$, which *damps* the prediction to correct the overshoot.
2. **Inherent Stability:** The $\tanh$ function naturally bounds the correction, ensuring that $D_t$ stays within a stable range of $(1 - a_1, 1 + a_1)$. This prevents the denominator from approaching zero or becoming negative, which would lead to numerical instability.
3. **Adaptive Strength:** The coefficient $a_1(t)$, still modulated by the normalized curvature magnitude $\tilde{\kappa}_t$ as in Section 4.2, controls the *strength* of the correction. This creates a two-level control: the sign of $\kappa_t$ determines the *direction* (boost/damp), while the magnitude $\tilde{\kappa}_t$ determines the *intensity* of the correction.

**Theoretical Justification.** This curvature-aware design fundamentally enhances our method's robustness. Instead of being a one-trick pony that only handles overshoots, CARP becomes an intelligent controller that actively corrects for both primary modes of extrapolation failure. The theoretical justification is no longer about a conditional error reduction under a specific assumption, but about a principled mechanism that uses a computable signal ($\kappa_t$) to approximate the direction of the error and apply a corrective action in the right direction. This ensures a more consistent reduction of error accumulation across a wider range of trajectory dynamics, making CARP broadly applicable and highly effective in low-step generation regimes.

### 4.4 EMPIRICAL VALIDATION: RESIDUAL PREDICTION YIELDS HIGHER FIDELITY

A core hypothesis of our work is that predicting residuals is a more stable and accurate strategy than directly predicting features for accelerated diffusion sampling. To provide direct, empirical evidence for this claim, we compare the fidelity of trajectories generated by two competing extrapolation methods:

- **Feature-based Prediction (Baseline):** Directly extrapolates the next feature map, i.e., $\hat{\mathbf{x}}_t = \text{Predictor}(\mathbf{x}_{t+1}, \mathbf{x}_{t+2}, \ldots)$.
- **Residual-based Prediction (Ours):** Extrapolates the next residual $\hat{\mathbf{r}}_t$ and then computes the feature map $\hat{\mathbf{x}}_t$ using the DDIM update rule with $\hat{\mathbf{r}}_t$.

We compare both methods against the ground truth trajectory generated with a full, non-accelerated forward pass.

**Quantitative Per-Step Fidelity.** First, we analyze the per-step accuracy of the generated features. For each step $t$, we measure the cosine similarity between the features generated by the prediction methods ($\hat{\mathbf{x}}_t$) and the ground truth features ($\mathbf{x}_t$). A higher similarity indicates a more accurate prediction and lower error accumulation.

Figure 3(a) plots this similarity over the denoising process. The results are unequivocal: the trajectory generated via **residual-based prediction consistently maintains a higher similarity** to the ground truth. This demonstrates that our chosen strategy leads to more accurate step-by-step predictions and effectively mitigates the accumulation of errors that plagues direct feature extrapolation.

**Qualitative Global Trajectory Structure.** Beyond per-step accuracy, it is crucial to maintain the global geometric structure of the generation trajectory. Drastic deviations from the true trajectory manifold can lead to significant artifacts in the final image. To visualize this, we use Principal

Component Analysis (PCA) to project the entire feature trajectories (from all timesteps) into a 2D space.

Figure 3(b) offers a striking visual confirmation of our method's superiority. The manifold traced by the **residual-based prediction** method closely mirrors the shape, curvature, and progression of the ground truth trajectory. In stark contrast, the trajectory from **feature-based prediction** quickly deviates and follows a significantly different path, indicating that its accumulated errors have distorted the fundamental generation process.

These empirical results provide compelling validation for our core design choice. Predicting residuals is not merely an alternative but a fundamentally more robust approach. It yields higher per-step fidelity (higher similarity) and better preserves the global structure of the generation manifold (closer PCA projection), both of which are critical for achieving high-quality results in accelerated sampling regimes.

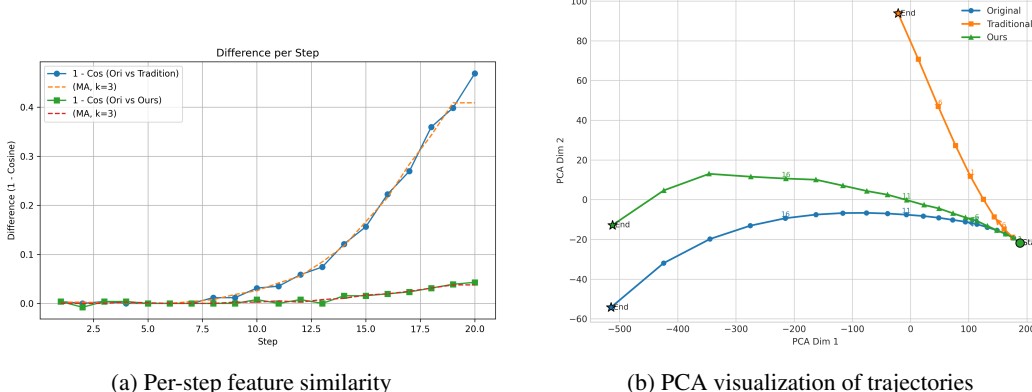

(a) Per-step feature similarity                    (b) PCA visualization of trajectories

Figure 3: Empirical comparison of feature-based vs. residual-based prediction fidelity. (a) Cosine similarity between predicted and ground truth features at each step. Our residual-based method (green) consistently achieves higher similarity than the feature-based baseline (blue). (b) 2D PCA projection of the entire trajectories. The path generated by our method closely follows the ground truth, while the baseline deviates significantly.

## 5 EXPERIMENTS

### 5.1 EXPERIMENTAL SETUP

**Model.** We evaluate our approach across three representative generative tasks: text-to-image, class-conditional image generation, and text-to-video. For text-to-image, we adopt the FLUX.1-dev model (Labs, 2024); for class-conditional generation, we use DiT-XL/2 (Peebles & Xie, 2022) on ImageNet; and for text-to-video, we employ Wan2.1 Wan et al. (2025). To ensure consistency, all models are standardized to 20 denoising steps, and all experiments are conducted on NVIDIA L40S GPUs under identical hardware settings.

**Evaluation Metrics.** For evaluation, we follow established benchmarks and datasets specific to each task. In text-to-image generation, we randomly sample 50k prompts from the COCO2017 (Lin et al., 2015) training set to produce $1024 \times 1024$ images, and further include 200 prompts from the DrawBench benchmark (Saharia et al., 2022) for supplementary qualitative comparison. Additionally, we conducted a user study for a more accurate validation of the method's effectiveness. For class-conditional image generation, we generate 50 samples per class at $256 \times 256$ resolution on the ImageNet dataset using the standard evaluation protocol. For text-to-video generation, we adopt VBench2(Zheng et al., 2025) as the benchmark and evaluate video synthesis performance under the same inference protocol. To assess performance, we report both *efficiency metrics* (FLOPs and inference latency) and *quality metrics*. For text-to-image generation, we evaluate FID, CLIP score, PickScore, Aesthetic score and Image Reward (Xu et al., 2023). For class-conditional image generation, we use FID, Inception Score (IS), Precision, and Recall.

Table 1: Quantitative results for **text-to-image** generation on the **50k COCO2017 training set**. Higher is better for quality metrics, and lower is better for efficiency metrics. "Aes." denotes Aesthetic Score. "PICK." denotes PICK Score. The best results are in **bold**, and the second best are underlined. Values marked with † indicate severe degradation in output image quality, with results falling outside the acceptable range for meaningful comparison.

| COCO2017 | Acceleration | | Image Quality | | | | |
|---|---|---|---|---|---|---|---|
| | Latency (s) ↓ | FLOPs (T) ↓ | FID ↓ | CLIP ↑ | PICK ↑ | Aes ↑ | User-Study |
| Flux.1[dev], 20steps | 12.11(1.00×) | 1487.80(1.00×) | 23.38 | 32.10 | - | 6.25 | - |
| ToCa ($\mathcal{N} = 5$) | 6.68(1.81×) | 509.48(2.92×) | 24.18 | 31.48 | 0.383 | 5.58 | 15.0% |
| $\Delta$-DiT ($\mathcal{N} = 3$) | 6.74(1.80×) | 694.54(2.14×) | 24.03 | 31.00 | 0.329 | 5.70 | 18.5% |
| TeaCache(Slow) | 8.52(1.42×) | 982.45(1.51×) | 23.90 | 31.38 | 0.424 | 6.03 | - |
| TeaCache(Fast) | 5.63(2.15×) | 610.60(2.44×) | 24.11 | 31.50 | 0.360 | 5.85 | 13.5% |
| TaylorSeer ($\mathcal{N} = 5, \mathcal{O} = 2$) | 5.22(2.31×) | 461.96(3.22×) | † | 31.52 | 0.311 | 4.95 | 7.0% |
| TaylorSeer ($\mathcal{N} = 6, \mathcal{O} = 2$) | 4.68(2.59×) | **387.59(3.84×)** | † | 30.95 | 0.252 | 4.46 | - |
| Ours(Slow) | 5.51(2.20×) | 582.60(2.55×) | **23.85** | **31.90** | **0.437** | **6.16** | - |
| Ours(Fast) | **4.20(2.88×)** | 506.23(2.94×) | 24.14 | **31.82** | 0.424 | 6.11 | **45.0%** |

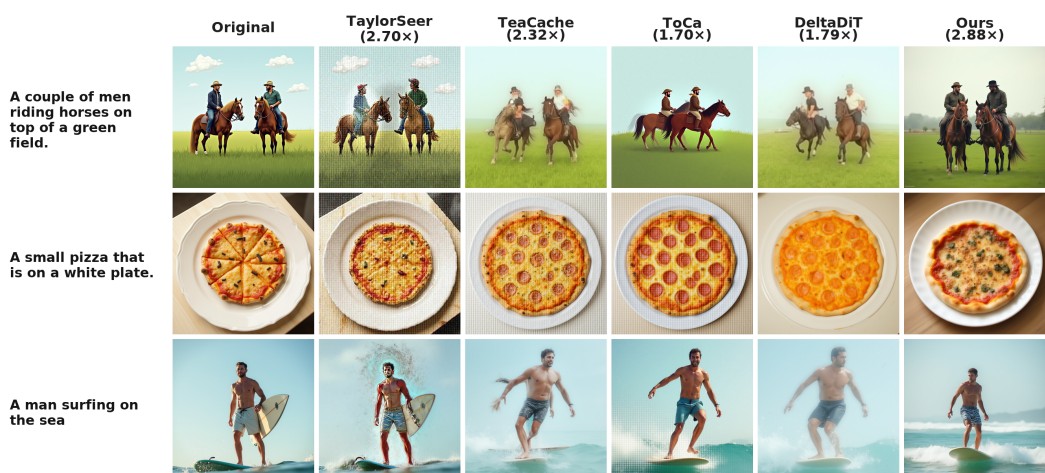

Figure 4: Qualitative comparison of different caching methods on FLUX. Each column corresponds to a distinct method, including Original, TaylorSeer, TeaCache, ToCa, DeltaDiT, and CARP (Ours), while each row represents a separate prompt.

## 5.2 RESULTS ON TEXT-TO-IMAGE GENERATION.

**Quantitative Study.** The qualitative results on the 50k COCO2017 training set are reported in Table 1. We adopt ToCa( Zou et al. (2024)), $\Delta$-DiT( Chen et al. (2024)), TeaCache( Liu et al. (2024)), and TaylorSeers( Liu et al. (2025b)) as baselines. For these methods, we executed their publicly available source code and selected the optimal hyperparameters to ensure a fair comparison. As shown in the table, our CARP achieves consistently superior performance in teCARP of both acceleration and image quality. In particular, CARP attains a $2.88\times$ speedup within only 20 timesteps, while better preserving the fidelity of the generated images. We provide relevant experimental results on Drawbench in the Appendix A.4.

**Qualitative Study.** Figure 4 compares visual quality in the few-step regime. CARP attains the largest speedup among baselines while incurring only minor perceptual degradation. For the prompt *"A man surfing on the sea."*, competing cache-based methods exhibit structural failures around the man's aCARP (e.g., collapse/fragmentation), whereas CARP preserves limb continuity and object contours. In the other two examples, alternative methods introduce grid-like (checkerboard) artifacts and inconsistent shading, while CARP maintains coherent illumination, textures, and scene geometry. These observations align with the reported quantitative results.

## 5.3 RESULTS ON CLASS-CONDITIONAL IMAGE GENERATION

Please refer to Appendix A.4 for more results on class conditional image generation.

## 5.4 RESULTS ON TEXT TO VIDEO GENERATION

Table 2: Quantitative comparison on text-to-video generation for Wan2.1 on VBench2.

| Method | Acceleration | | | | VBench2 Score |
|---|---|---|---|---|---|
| | Latency (s)↓ | Speed ↑ | FLOPs (T)↓ | Speed ↑ | |
| Wan2.1, 20steps | 88 | 1.00× | 3568.83 | 1.00× | 64.16% |
| Teacache(Slow) | 75 | 1.17× | 3076.39 | 1.16× | **60.73%** |
| Teacache(Fast) | 61 | 1.44× | 2583.95 | 1.28× | 58.40% |
| TaylorSeer ($\mathcal{N} = 3, \mathcal{O} = 1$) | 67 | 1.31× | 1954.23 | 1.83× | 54.74% |
| TaylorSeer ($\mathcal{N} = 4, \mathcal{O} = 1$) | 53 | 1.66× | 1876.24 | 1.90× | 54.50% |
| Ours(Fast) | **51** | **1.72×** | 2055.24 | 1.74× | 60.38% |

**Quantitative Study.** On Wan2.1, our CARP delivers the *best acceleration* with **51 s** latency (**1.72** ×) and **2055.24 T** FLOPs (**1.74** ×), while maintaining a VBench2 score of 46.13%—only **2.51** points below the 20-step baseline (48.64%). This places CARP on the Pareto frontier: it attains the largest speedup with minimal quality drop, whereas alternative caching/prediction approaches either achieve smaller gains or incur noticeably higher degradation, especially in the $> 1.5\times$ regime.

**Qualitative Study.** Figure 11 demonstrates the exceptional capability of our method in accelerating Wan2.1 inference while preserving video quality. The videos processed through our approach exhibit minimal degradation, maintaining high visual fidelity comparable to the original outputs. This visual preservation, achieved under significant computational speedup, underscores the effectiveness of our residual prediction framework in maintaining temporal coherence and structural integrity across frames. The qualitative results align with our quantitative metrics, confirming that our acceleration method does not compromise the perceptual quality of the generated video content.

## 5.5 ABLATION STUDIES

We conduct a comprehensive ablation study to evaluate the individual contributions of CARP components on Flux along three axes: (i) the curvature threshold N governing prediction triggering, (ii) the degree of feature utilization in the rational function's numerator, and (iii) the granularity of prediction application (single block stack, dual block stack, full stack, or vector fields). Results in Table 6 on Appendix A.5 show that CARP remains robust across hyperparameter variations: smaller N increases prediction frequency but may introduce noise, while higher degrees of feature utilization in the rational function's numerator improve approximation at an increased computational cost. Applying **PREDICT** to the full DiT stack yields the highest speedup with maintained quality, whereas single-block acceleration is limited. Our optimal configuration uses N=1.4(almost all subsequent steps use prediction), first-order predictor, and full-stack application, achieving the best speed–quality trade-off with negligible degradation. This setup is adopted as default in all subsequent experiments. We also conducted experiments without the denominator to validate the role of curvature in rational function prediction. The results confirmed that incorporating curvature into the denominator effectively guides the linear regression prediction.

## 6 CONCLUSION

We proposed CARP, a novel training-free method for accelerating Diffusion Transformers under low denoising steps. By shifting the prediction target to stable residual updates and employing a curvature-aware rational extrapolator, CARP effectively overcomes the error accumulation and instability that limit existing caching methods. Extensive experiments demonstrate that our approach achieves significant speedups while maintaining high output quality across multiple generative tasks. This work highlights residual prediction as a robust and efficient paradigm for diffusion model acceleration.

## 7 REPRODUCIBILITY STATEMENT

The empirical analyses and the theoretical error analysis of our prediction method presented in this paper are fully reproducible. We have explicitly considered all the necessary assumptions during the derivation process to ensure clarity and reproducibility. Additionally, we provide detailed information about the model parameters and GPU configurations used in our experiments. We believe that our algorithm can be independently reproduced by other researchers, as all necessary information has been included for transparency and replication.

## 8 ETHICS STATEMENT

The empirical analyses and the theoretical error analysis of our prediction method presented in this paper are fully reproducible. We have explicitly considered all the necessary assumptions during the derivation process to ensure clarity and reproducibility. Additionally, we provide detailed information about the model parameters and GPU configurations used in our experiments. We believe that our algorithm can be independently reproduced by other researchers, as all necessary information has been included for transparency and replication.

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

# A APPENDIX

## A.1 LLM USAGE

During the writing of this paper, Large Language Models (LLMs) were utilized exclusively for language-related tasks. Specifically, LLMs were employed to assist in translating and polishing the text, with the primary goal of improving the fluency and readability of the writing and reducing grammatical inaccuracies.

It is important to note that LLMs were not used to generate core ideas, formulate research methodologies, derive conclusions, or produce any of the key code implementations presented in this work. All intellectual contributions, analytical reasoning, and critical technical content remain entirely our own. The use of LLMs was strictly limited to enhancing the linguistic quality of the final manuscript.

## A.2 DETAILED ALGORITHM PROCESS

---

**Algorithm 1:** CURVATURE-AWARE RESIDUAL PREDICTION

---

**Input:** initial noise $\mathbf{x}_T$, model $\theta$, warmup steps $P$, curvature threshold $N$
**Output:** generated sample $\mathbf{x}_0$

**Warmup (standard denoising; collect residuals): for** $t \leftarrow T$ **to** $T - P + 1$ **do**
  | $\mathbf{r}_t \leftarrow \text{TransformerResidual}(\mathbf{x}_t, \theta)$ ;      // $\mathbf{r}_t = \mathcal{T}_\theta(\mathbf{x}_t) - \mathbf{x}_t$
  | $\mathbf{y}_t \leftarrow \mathbf{x}_t + \mathbf{r}_t$ ;      // Transformer block output
  | $\mathbf{x}_{t-1} \leftarrow \text{SchedulerStep}(\mathbf{x}_t, \mathbf{y}_t, t)$
  | Append $\mathbf{r}_t$ to *history*

**Accelerate (predict or recompute per curvature): for** $t \leftarrow T - P$ **to** $1$ **do**
  | Estimate curvature $\kappa_t$ from recent residuals in *history* (e.g., $\{\mathbf{r}_{t+2}, \mathbf{r}_{t+1}, \mathbf{r}_t\}$);
  | **if** $\kappa_t < N$ **then**
    |    // Low curvature: use rational predictor to predict the residual
    | $\hat{\mathbf{r}}_t \leftarrow \text{RationalPredict}(\mathbf{r}_{t+1}, \mathbf{r}_{t+2}, \mathbf{r}_{t+3})$ ;      // Use rational function-based prediction
    | $\mathbf{y}_t \leftarrow \mathbf{x}_t + \hat{\mathbf{r}}_t$ ;      // use predicted residual as block output
    | $\mathbf{x}_{t-1} \leftarrow \text{SchedulerStep}(\mathbf{x}_t, \mathbf{y}_t, t)$ Append $\hat{\mathbf{r}}_t$ to *history*
  | **else**
    |    // High curvature: compute true residual via Transformer
    | $\mathbf{r}_t \leftarrow \text{TransformerResidual}(\mathbf{x}_t, \theta)$ $\mathbf{y}_t \leftarrow \mathbf{x}_t + \mathbf{r}_t$ $\mathbf{x}_{t-1} \leftarrow \text{SchedulerStep}(\mathbf{x}_t, \mathbf{y}_t, t)$
    |    Append $\mathbf{r}_t$ to *history*

**return** $\mathbf{x}_0$

---

## A.3 RELATED WORK

**Acceleration of diffusion models.** As diffusion models (Ho et al., 2020) increasingly pursue scalability, the model sizes continue to grow, leading to a corresponding rise in research focused on accelerating diffusion models to alleviate the problem of poor real-time performance. Current acceleration techniques for diffusion models can be categorized into three main directions: First, similar to traditional network lightweight techniques, numerous approaches have focused on pruning (Fang et al., 2023; Zhu et al., 2024), quantization (Kim et al., 2025; Li et al., 2023a; Shang et al., 2023), and distillation (Li et al., 2023b) of noise estimation networks to achieve a smaller model that retains comparable performance. second, many efforts have been made to reduce the number of denoising steps. Techniques such as DDIM (Song et al., 2020a) has reduced the number of denoising steps required by the model, enabling it to achieve excellent sampling results with fewer steps. Additionally, some approaches focus on more efficient ODE or SDE solvers (Song et al., 2020b; Liu et al., 2022), which allow for a reduction in the number of sampling steps while maintaining the quality of the generated results. Finally, other methods (Chen et al., 2024; Ma et al., 2024; Selvaraju et al., 2024) reuse intermediate features between consecutive time steps to avoid redundant computations, thereby enhancing sampling efficiency.

**Cache Mechanism.** In diffusion models, caching mechanisms exploit the high temporal similarity of features between adjacent denoising steps. By storing and reusing feature maps computed in previous steps, these methods significantly reduce redundant computations, thereby lowering computational overhead and accelerating inference. The original concept of feature caching was primarily designed for U-Net architectures, leveraging their skip connections to efficiently propagate and reuse multi-level features. Methods such as Faster Diffusion (Li et al., 2024) and DeepCache (Ma et al., 2024) focus on caching the features by outputs of specific U-Net blocks. However, these are designed specifically for U-Net architectures and cannot be directly applied to modern Diffusion Transformer (DiT) models. While DiT enhances scalability, it also introduces significant computational overhead, leading to an increase in computational costs. Advanced techniques such as FoRA (Selvaraju et al., 2024) and PAB (Zhao et al., 2024) leverage attention and MLP representation reuse, while $\Delta$-DiT (Chen et al., 2024) and BlockDance (Zhang et al., 2025) focus on reusing block features to skip the computation of certain blocks. ToCa (Zou et al., 2024) and Tokencache (Lou et al., 2024) achieves effective acceleration by innovatively shifting the caching target to tokens, thereby reducing information loss. TeaCache (Liu et al., 2025a) predicts output change and gate reuse, by utilizing the differences in the noise input through time-step embedding. Recent innovative research includes TaylorSeers (Liu et al., 2025c), which uses Taylor expansion to approximate the denoising trajectory and predict the features for the next time step.

A.4 More Quantitative Results

**Results on class-conditional image generation.** The results of the 50k ImageNet images are shown in Table 3. We extend ToCa, $\Delta$DiT, FORA+GOC (Qiu et al., 2025) and TaylorSeer to DiT/XL-2 as baselines, demonstrating that CARP significantly outperfoCARP the others in both acceleration ratio and generation quality. CARP achieves an FID-50k of

Table 3: Quantitative comparison on class-to-image generation on ImageNet with DiT-XL/2.

| Imagenet | Acceleration | | | | Image Quality | | | |
|---|---|---|---|---|---|---|---|---|
| | Latency (s)↓ | Speed↑ | FLOPs (T)↓ | Speed↑ | FID↓ | IS↑ | Precision↑ | Recall↑ |
| DiT-XL/2, 20steps | 1.71 | 1.00× | 9.49 | 1.00× | 3.56 | 221.27 | 0.78 | 0.58 |
| ToCa(N = 3) | 1.26 | 1.35× | 4.01 | 2.37× | 10.72 | 164.40 | 0.69 | 0.49 |
| $\Delta$-DiT (N = 3) | 1.31 | 1.31× | 6.43 | 1.48× | 8.86 | 170.96 | 0.70 | 0.55 |
| FORA+GOC(Cache=50%) | 1.19 | 1.44× | 5.93 | 1.60× | 6.53 | 193.51 | 0.74 | 0.53 |
| TaylorSeer ($\mathcal{N} = 4, \mathcal{O} = 3$) | **1.13** | **1.51×** | **2.85** | **3.32×** | 7.86 | 175.11 | 0.72 | 0.54 |
| TaylorSeer ($\mathcal{N} = 3, \mathcal{O} = 2$) | 1.19 | 1.44× | 3.80 | 2.49× | 7.84 | 175.99 | 0.71 | 0.53 |
| Ours(Fast) | 1.17 | 1.46× | 5.79 | 1.64× | 6.93 | 185.12 | 0.72 | 0.54 |

Table 4: Ablation study on the impact of prediction granularity, order of predictor, and curvature threshold $N$ on inference efficiency and generation quality. The "Curvature-Aware" column indicates whether curvature is incorporated into the denominator to guide the prediction in the rational function form.

| Prediction Target | Curvature-Aware | Order | $N$ | Latency (s) | Aes ↑ | CLIP ↑ | Image Reward ↑ |
|---|---|---|---|---|---|---|---|
| FullDiTBlock | ✔ | 1 | 1.4 | 4.20 (2.88×) | 5.76 | 31.83 | 0.9184 |
| | | | 1.0 | 5.50(2.20×) | 5.77 | 31.97 | 0.9236 |
| | | | 0.8 | 7.53( 1.63×) | 5.80 | 32.02 | 0.9562 |
| Vector Fields | ✔ | 1 | 1.4 | 4.01(3.01×) | 5.67 | 31.61 | 0.8619 |
| FullDiTBlock | ✔ | 2 | 1.4 | 5.24(2.31×) | 5.68 | 31.38 | 0.8687 |
| DualDiTBlock | ✔ | 1 | 1.4 | 8.90(1.36×) | 5.10 | 31.31 | 0.7921 |
| SingleDiTBlock | | | | 6.24(1.94×) | 5.69 | 31.66 | 0.8717 |
| FullDiTBlock | ✘ | 1 | 1.4 | 4.32(2.80×) | 5.72 | 31.36 | 0.8149 |

**Quantitative results for text-to-image generation on DrawBench.** The DrawBench results show that CARP delivers the strongest overall quality among caching methods: it achieves the best Image Reward (0.91) and best Aesthetic score (5.76), while tying for second-best PickScore (0.47). Tea-Cache attains the top PickScore (0.48) but lags on the other metrics. $\Delta$-DiT and ToCa trail across

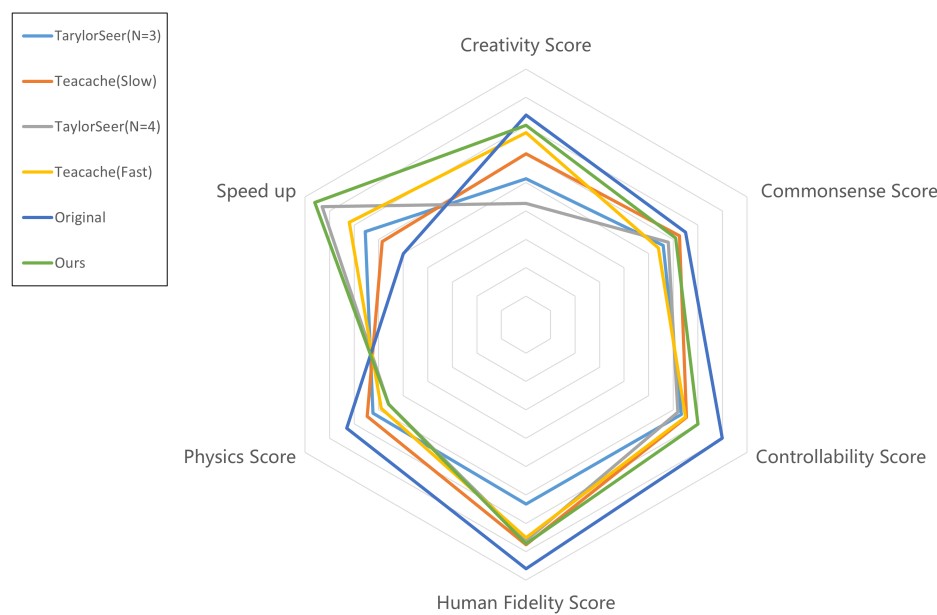

Figure 5: VBench metrics and acceleration ratio of proposed CARP and other methods.

metrics, and polynomial extrapolation (TaylorSeer) degrades quality notably. Overall, our curvature-aware residual prediction maintains higher perceptual quality than prior caching approaches and narrows the gap to the original baseline.

Table 5: Quantitative results for **text-to-image** generation on **DrawBench**. Higher is better for quality metrics, and lower is better for efficiency metrics. The best results are in **bold**, and the second best are underlined.

| DrawBench | Image Quality | | |
|---|---|---|---|
| | PickScore ↑ | Image Reward ↑ | Aes ↑ |
| Flux.1[dev], 20steps | – | 1.01 | 5.83 |
| $\Delta$-DiT (N = 3) | 0.41 | 0.52 | 5.42 |
| ToCa(N=5) | 0.44 | 0.82 | 5.24 |
| TeaCache($\delta = 0.25$) | **0.48** | 0.85 | 5.73 |
| TeaCache($\delta = 0.4$) | 0.47 | 0.86 | 5.71 |
| TeaCache($\delta = 0.6$) | 0.45 | 0.74 | 5.62 |
| TaylorSeer ($\mathcal{N} = 5, \mathcal{O} = 2$) | 0.38 | 0.69 | 4.75 |
| TaylorSeer ($\mathcal{N} = 6, \mathcal{O} = 2$) | 0.32 | 0.68 | 4.40 |
| Ours(Fast) | 0.47 | **0.91** | **5.76** |

**Text-to-Video Generation.** Please refer to Figure 5

**User Study on Text-to-Image among different methods on FLUX.** We conducted a user study on text-to-image generation using FLUX, involving 50 volunteers who were asked to compare images and their corresponding prompts across various caching methods. Participants were tasked with selecting the method that retained the highest image quality. Importantly, to ensure fairness, the volunteers were unaware of which specific method generated each image, ensuring unbiased comparisons. The result is shown in Table 6.

A.5    ABLATION RESULTS.

Please refer to Tabel 6.

Table 6: Ablation study on the impact of prediction granularity, order of predictor, and curvature threshold $N$ on inference efficiency and generation quality. The "Curvature-Aware" column indicates whether curvature is incorporated into the denominator to guide the prediction in the rational function form.

| Prediction Target | Curvature-Aware | Order | $N$ | Latency (s) | Aes ↑ | CLIP ↑ | Image Reward ↑ |
|---|---|---|---|---|---|---|---|
| FullDiTBlock | ✔ | 1 | 1.4 | 4.20 (2.88×) | 5.76 | 31.83 | 0.9184 |
| | | | 1.0 | 5.50 (2.20×) | 5.77 | 31.97 | 0.9236 |
| | | | 0.8 | 7.53 ( 1.63×) | 5.80 | 32.02 | 0.9562 |
| Vector Fields | ✔ | 1 | 1.4 | 4.01(3.01×) | 5.67 | 31.61 | 0.8619 |
| FullDiTBlock | ✔ | 2 | 1.4 | 5.24(2.31×) | 5.68 | 31.38 | 0.8687 |
| DualDiTBlock SingleDiTBlock | ✔ | 1 | 1.4 | 8.90(1.36×) 6.24(1.94×) | 5.10 5.69 | 31.31 31.66 | 0.7921 0.8717 |
| FullDiTBlock | ✗ | 1 | 1.4 | 4.32(2.80×) | 5.72 | 31.36 | 0.8149 |

## A.6 MORE QUALITATIVE RESULTS

This section we show more qualitative results between several cache methods.

**Text-to-Image Generation**

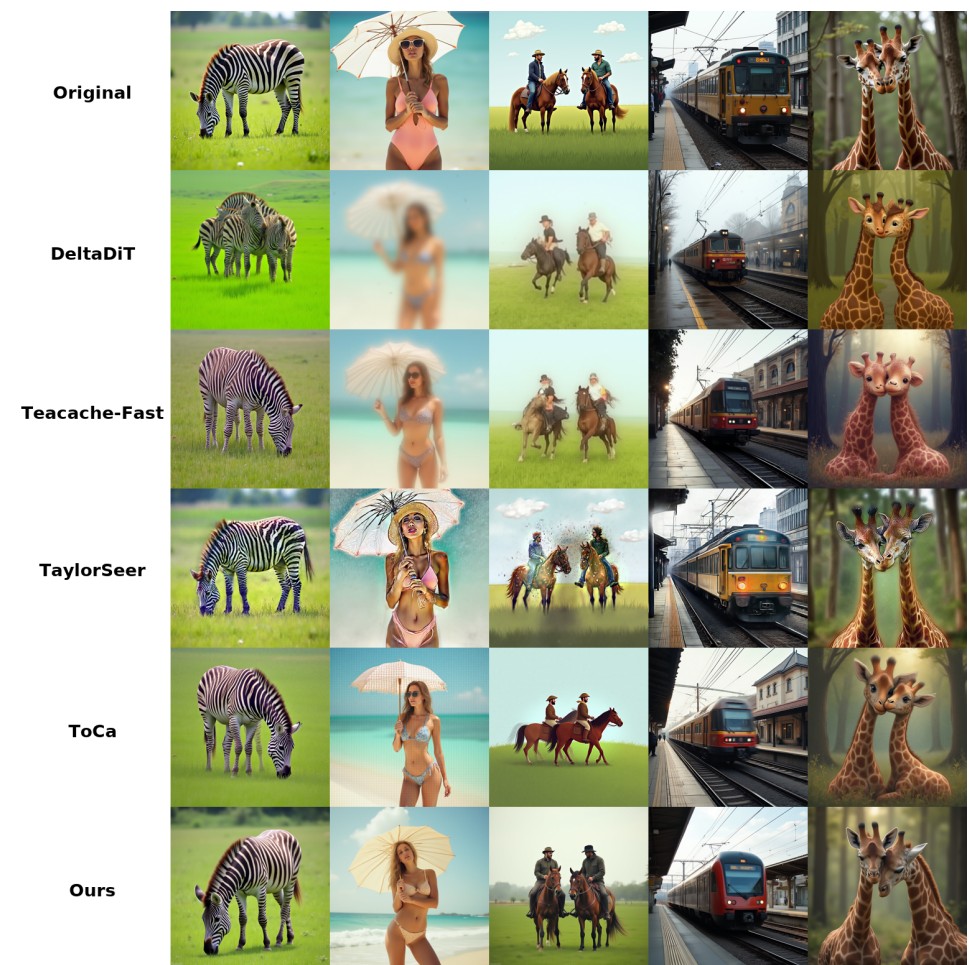

Figure 6: More Results of text-to-image task on FLUX

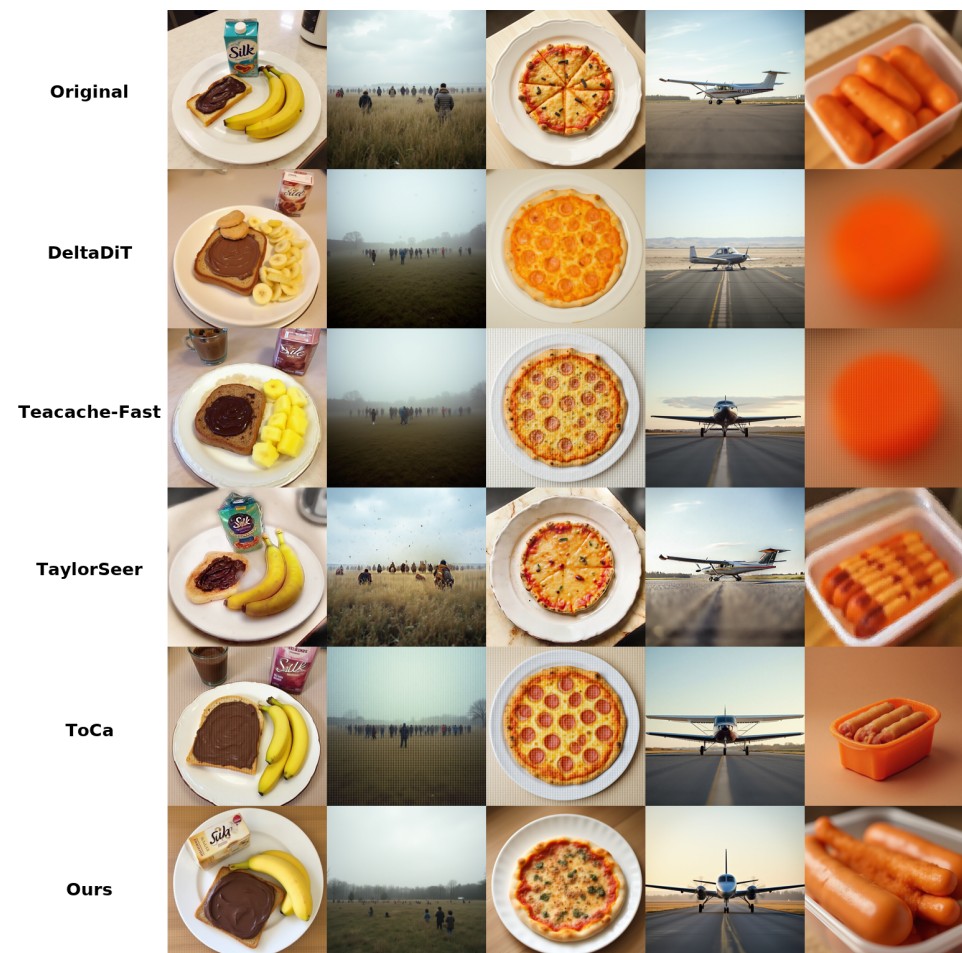

Figure 7: More Results of text-to-image task on FLUX

**Text-to-Video Generation**

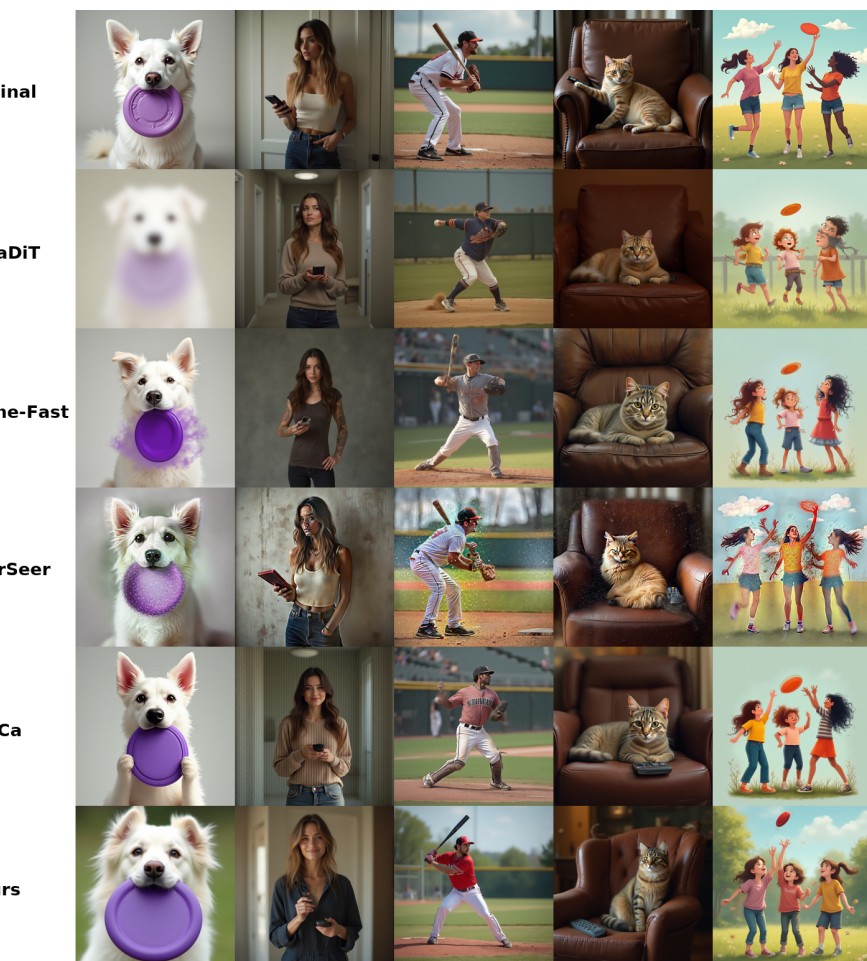

Figure 8: More Results of text-to-image task on FLUX

972
973
974
975
976
977
978
979
980
981
982
983
984
985
986
987
988
989
990
991
992
993
994
995
996
997
998
999
1000
1001
1002
1003
1004
1005
1006
1007
1008
1009
1010
1011
1012
1013
1014
1015
1016
1017
1018
1019
1020
1021
1022
1023
1024
1025

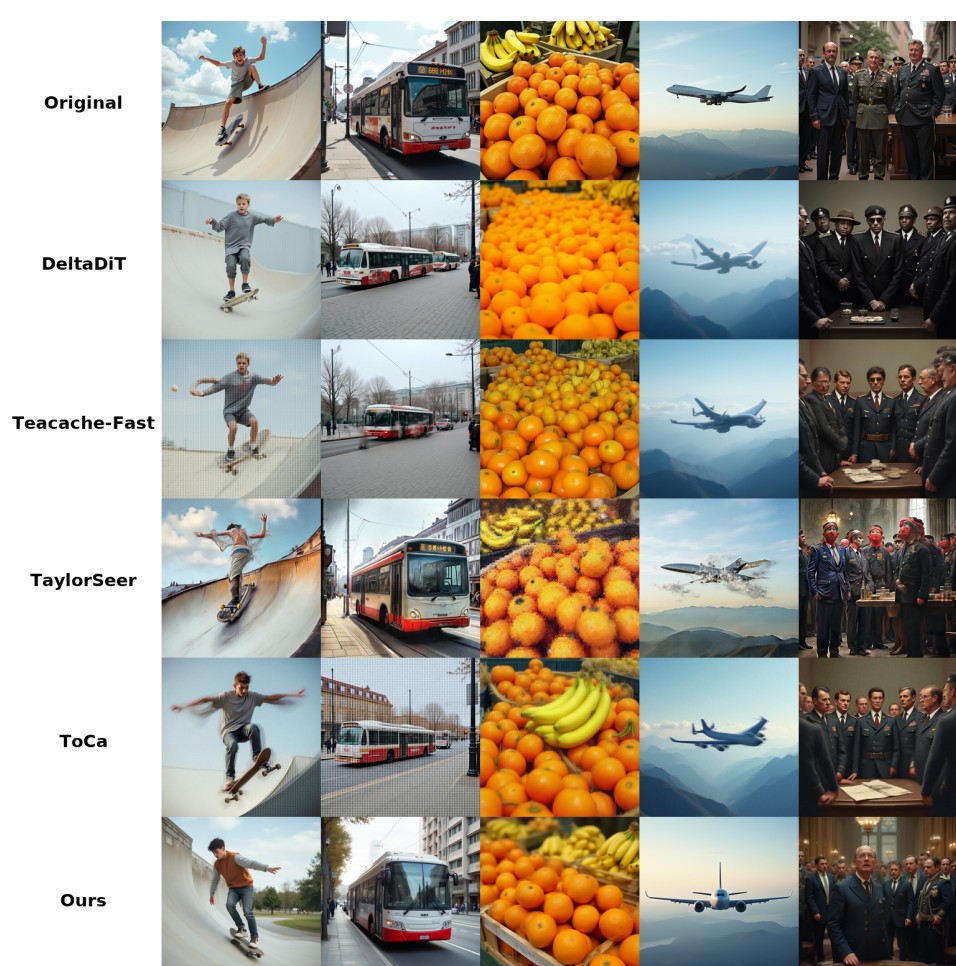

Figure 9: More Results of text-to-image task on FLUX

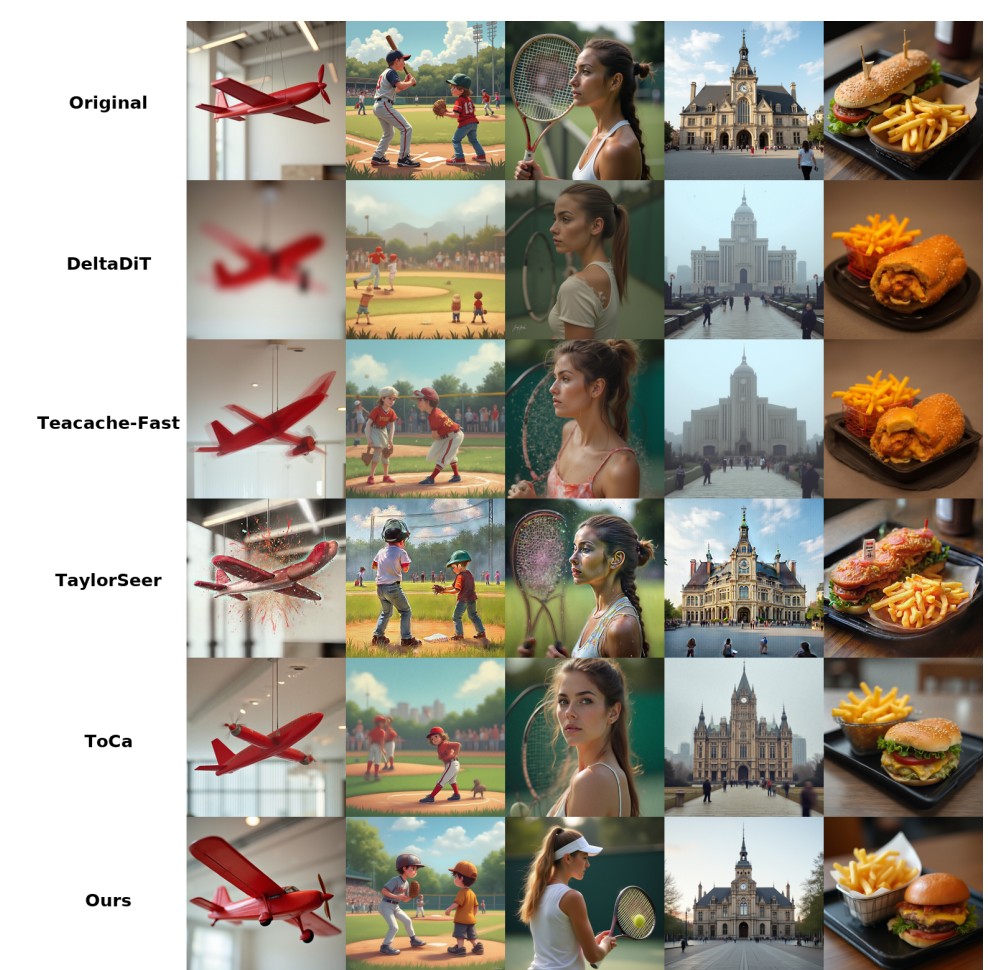

Figure 10: More Results of text-to-image task on FLUX

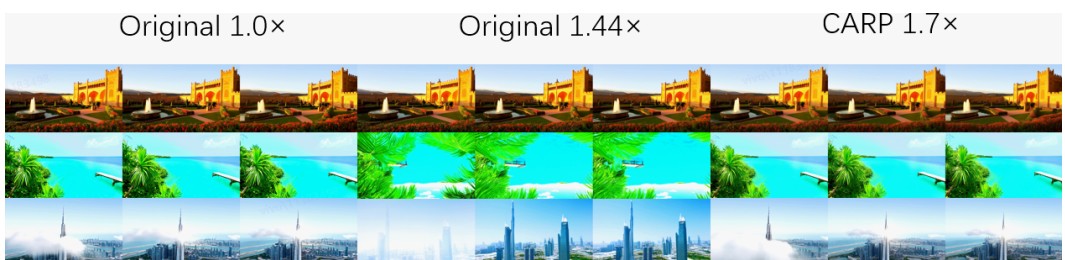

Figure 11: Enter Caption

