# OpenReview forum: "Curvature-Aware Residual Prediction for Stable and Faithful Diffusion Transformer Acceleration Under Large Sampling Intervals"
_ICLR.cc/2026/Conference — Submitted to ICLR 2026_

### Official Review · Reviewer_1vXq · 2025-10-26

**Soundness:** 3
**Presentation:** 3
**Contribution:** 3
**Rating:** 4
**Confidence:** 5

**Summary:**

This paper proposes the CARP: a training-free, curvature-aware residual prediction scheme that ensures theoretical superiority over current prediction-based methods and reuse-based acceleration methods by shifting the prediction target from raw features to residual updates within the Diffusion Transformer Blocks. The author introduces a rational-function-based prediction method that involves a numerator performing linear extrapolation using adjacent features and a denominator that incorporates discrete curvature to ensure optimal skipping in steps. Experiment results show strong speedups with preserved qualities of generation in both image and video generation tasks.

**Strengths:**

1. Practical deployability. CARP is training-free and plug-and-play with no architecture changes. There is also minimal overhead introduced in both memory and inference, given the method uses a tiny, fixed history (3 residuals) and a simple gate, so it's easy to integrate and adds little overhead while yielding meaningful speedups.

2. Stability under large strides. The curvature-gated fallback with rational predictor adaptively controls the error introduced in acceleration, avoiding over-/undershoot and reducing error accumulation in comparison with evaluated baselines.

3. Methodology novelty. While many existing studies have studied the step-wise sparsity in multi-step generation, the method in this paper provides a new solution with mathematical justification.

**Weaknesses:**

1. Limited Ablation on Samplers. It appears that the DDIM is the only sampler evaluated; please provide a solver sweep (DPM-Solver / DPM-Solver++, Euler, and a flow-matching ODE case) under identical prompts, seeds, and step counts (e.g., 20/50), reporting FID and LPIPS, latency, and speedups to verify solver-agnostic robustness.

2. Missing Baselines. There are many similar step-wise sparsity-aware open-sourced baselines that have been proposed before this paper but not yet evaluated: DeepCache (CVPR 24'), AdaptiveDiffusion (NeurIPS 24'), SADA (ICML 25'), AB-Cache (ArXiv, optional as may not be open-sourced yet, but good to mention results in paper). Please evaluate these baselines on one of the DDIM/DPM Solver/Euler samplers. Please also provide all hyperparameters used in the comparison of these open-source baselines.

3. Missing Metric. The author should also report the LPIPS (also reported in TaylorSeer but missing in CARP) as a metric to justify the perceptual distance shifting from the unaccelerated model. Please compare your method with the TaylorSeer and baselines mentioned above on FID, PSNR, SSIM, and LPIPS with speedups.



Refs:

[1] Narayanan, Arvind, et al. "Deepcache: A deep learning based framework for content caching." Proceedings of the 2018 Workshop on Network Meets AI & ML. 2018.

[2] Ye, Hancheng, et al. "Training-free adaptive diffusion with bounded difference approximation strategy." Advances in Neural Information Processing Systems 37 (2024): 306-332.

[3] Jiang, Ting, et al. "Sada: Stability-guided adaptive diffusion acceleration." arXiv preprint arXiv:2507.17135 (2025).

[4] Yu, Zichao, et al. "AB-Cache: Training-Free Acceleration of Diffusion Models via Adams-Bashforth Cached Feature Reuse." arXiv preprint arXiv:2504.10540 (2025).

**Questions:**

1. The author states, "selected the optimal hyperparameters to ensure a fair comparison" in the experiment section. Can the author list all hyperparameters for each baseline they have tested?

---

### Official Review · Reviewer_sQVm · 2025-10-29

**Soundness:** 3
**Presentation:** 2
**Contribution:** 3
**Rating:** 6
**Confidence:** 4

**Summary:**

The paper proposes a training-free acceleration framework for Diffusion Transformers. Instead of reducing the number of denoising steps, CARP predicts the residual updates between steps to avoid redundant Transformer computations. It introduces a rational-function predictor that extrapolates residuals using recent history. A curvature-based gating mechanism adaptively decides whether to use the predicted residual or perform a real forward pass, ensuring both stability and fidelity across smooth and complex denoising regimes. CARP achieves up to 2.9× speedup on FLUX and 1.5–1.7× on DiT-XL/2 and Wan 2.1 with minimal quality degradation.

**Strengths:**

1. The method offers a training-free, plug-and-play acceleration solution that can be integrated into any Diffusion Transformer without retraining or architecture modification.

2. The paper provides strong empirical and theoretical validation. Experiments on multiple high-end Diffusion Transformer backbones—FLUX, DiT-XL/2, and Wan2.1—demonstrate consistent acceleration (up to 2.9×) with minimal perceptual or quantitative degradation.

**Weaknesses:**

1. Limited evaluation scope and lack of scaling/generalization evidence.

The evaluation focuses exclusively on three Diffusion Transformer architectures (FLUX, DiT-XL/2, and Wan 2.1), each tested only under a 20-step denoising schedule. However, the curvature-aware predictor’s stability and error behavior could change under longer sampling horizons (e.g., 50 steps or 100 steps), where residual trajectories evolve more gradually but accumulate error differently. Assessing CARP’s performance across multiple step settings would provide stronger evidence of its robustness.

Moreover, although CARP claims to be architecture-agnostic, all experiments are conducted on Transformer-based diffusion models. It remains unclear how well the curvature-aware residual prediction generalizes to UNet-based diffusion models such as Stable Diffusion XL (SDXL) or EDM, where spatial convolutional dependencies differ substantially from DiT’s attention-driven dynamics. Including such baselines would better support the general-applicability claim.

2. Clarification on the Definition and Scope of Single, Dual, and Full DiTBlocks in Table 6.

I find the terminology in Table 6 — Single, Dual, and Full DiTBlocks — somewhat unclear. Does “Dual DiTBlock” refer to the double-stream blocks used in FLUX? What exactly does “Full DiTBlock” mean? Please clarify. Also, am I correct in understanding that Table 6 studies the performance difference between partial-block skipping and skipping the entire model stack at once?

3. Limited exploration of curvature threshold sensitivity.

The gating threshold 𝑁=1.4 is empirically selected but not deeply analyzed. Understanding how this hyperparameter trades off between stability and acceleration—and whether adaptive thresholds (learned or schedule-based) could outperform a fixed value—would make CARP more robust across models.

**Questions:**

I am curious about this training-free method: does it always skip the same block for different text inputs?

---

### Official Review · Reviewer_XAXC · 2025-10-29

**Soundness:** 1
**Presentation:** 1
**Contribution:** 2
**Rating:** 2
**Confidence:** 4

**Summary:**

CARP is a training-free, model-agnostic acceleration for Diffusion Transformers that predicts residual updates (instead of raw features) using a curvature-aware rational predictor with gating, reducing accumulated errors—especially at large sampling steps. It delivers up to 2.88×/1.46×/1.72× speedups on FLUX/DiT-XL/2/Wan2.1 at 20 steps while preserving quality (FID/CLIP/Aesthetic/VBench) and ~25% higher user preference on FLUX, outperforming cache- and Taylor-based baselines.

**Strengths:**

1. The proposed method achieves SOTA performance.
2. It is a training-free method that does not consume many resources.

**Weaknesses:**

1. The writing is very poor.
2. The reason for employing the residual instead of the output itself is not clear.
3. The meaning of $\widetilde\kappa_t$ is not well justified.
4. The reason for using the first-order residual when $\widetilde\kappa_t$ is large is also not clear.
5. I am curious about the memory consumption of this method, as it requires storing lots of residuals.
6. The performance on SOTA video generation methods, e.g., Hunyuan and Wan2.1 on high-resolution generation, e.g., 720p and beyond, is missing. The acceleration of more powerful video generation models towards higher resolution should be more challenging and practical.
7. The author should include the experiments on few-step diffusion models.
8. I am curious why the authors do not compare their method with TaylorSeer at a higher order compensation, which can achieve better performance.

**Questions:**

N/A

---

### Official Review · Reviewer_UBz3 · 2025-11-01

**Soundness:** 3
**Presentation:** 3
**Contribution:** 3
**Rating:** 4
**Confidence:** 4

**Summary:**

The paper proposes CARP, a training-free inference acceleration scheme for Diffusion Transformers. Instead of extrapolating raw hidden states across timesteps, CARP predicts end-to-end residual updates from a short history and uses a rational extrapolator whose denominator is modulated by a curvature signal; a curvature-guided gate decides whether to trust the prediction or fall back to a full forward pass. Experiments on FLUX (text-to-image), DiT-XL/2 (ImageNet), and Wan2.1 (text-to-video) report speedups up to 2.88× with limited degradation on standard metrics, plus a user preference boost on FLUX. The history window is fixed to 3 residuals and the method uses curvature thresholds for both the rational term and the hard gate. A theoretical note links the sign of discrete curvature to the direction of linear-extrapolation error, motivating the denominator’s form.

**Strengths:**

1. The overall idea (predict residuals, not features) and the rational predictor/gating are easy to follow.

2. CARP does not modify architectures and targets the low-step regime most relevant for latency.

3. Reported speedups at 20 steps on FLUX/DiT-XL/2/Wan2.1 with competitive quality (e.g., FLUX up to 2.88×) and a positive user study.

**Weaknesses:**

1. Heavy reliance on hand-crafted thresholds / hyperparameters. CARP’s gating and denominator strength hinge on manually chosen thresholds (e.g., 𝑁). While there is an ablation, the paper does not convincingly show that these can be set once and generalize across models, datasets, and samplers without per-scenario tuning.

2. Fixed and narrow temporal context. The method fixes the history window to 3, which may limit robustness when trajectories are noisier or when step schedules differ. There’s no exploration of adaptive or larger windows.

3. Decision signals feel manual/heuristic. The normalized curvature, thresholds, and hard gate are designed features rather than learned criteria; it’s unclear how stable they are under distribution shift (e.g., different prompt mixes, seeds, schedulers).

4. Missing comparisons to few-step/step-distillation baselines. Since the paper targets aggressive low-step regimes, it should compare against step-distilled or solver-distilled models under matched wall-clock/quality budgets, not only cache/prediction baselines (ToCa, TeaCache, Δ-DiT, TaylorSeer). The current suite does not settle the “best way” to achieve low-latency sampling.

5. Limited discussion of overheads. Computing curvature, gating, and rational inference adds control-flow and tensor ops. The paper would benefit from a breakdown of where the speedups come from (skipped blocks vs. cache reuse vs. prediction) and their sensitivity to GPU/TPU kernels.

**Questions:**

1. Window size/generalization. How does CARP perform with other window sizes or an adaptive window (e.g., expand when curvature is stable, shrink when volatile)? Please report sensitivity curves and costs.
2. Can you provide cross-model, cross-dataset results showing that a single set of hyper-parameter works well without retuning? Any automatic calibration procedure?
3. Please include a kernel-level latency breakdown (prediction/gating vs. DiT forward) to show that wins aren’t hardware-specific and to guide future optimizations.

---

### Meta-Review · Area_Chair_3fGK · 2026-01-01

**Summary:**

The paper proposes CARP (Curvature-Aware Residual Prediction), a training-free acceleration framework for Diffusion Transformers. The method shifts the prediction target from raw features to residual updates and employs a rational function-based predictor modulated by a curvature-guided gating mechanism. Experiments on FLUX, DiT-XL/2, and Wan2.1 demonstrate competitive speedups (up to 2.88x) with maintained quality metrics in low-step regimes.

However, the consensus among the reviewers is that the paper is not ready for publication. While the core idea of residual prediction is acknowledged as interesting, significant concerns remain regarding the robustness of the heuristics, the lack of comprehensive baselines (particularly against distillation methods), and clarity issues in the presentation. Crucially, the authors did not submit a rebuttal. Without a response to address the substantial questions raised regarding hyperparameter sensitivity, generalization across solvers, and comparative fairness, the concerns outlined below remain outstanding. Therefore, the AC recommends rejection.

**Reviewer Concerns:**

As no rebuttal was submitted, no reviewer concerns were addressed. The following concerns remain outstanding and form the basis of the rejection decision:

- Reliance on Heuristics and Hyperparameters:
   - Reviewers expressed strong concern over the heavy reliance on hand-crafted thresholds (e.g., curvature threshold N) and fixed window sizes (history of 3).


   - It remains unproven whether these hyperparameters generalize across different models, datasets, and schedulers without per-scenario tuning.
   - There is a lack of sensitivity analysis regarding these thresholds.


- Insufficient Baselines and Comparisons:
   - The paper fails to compare against step-distillation or solver-distillation baselines, which are critical competitors in the low-step regime the authors are targeting.


   - There is a lack of evaluation across different solvers (e.g., DPM-Solver++, Euler) to verify solver-agnostic robustness.
   - Missing comparisons to relevant open-source baselines such as DeepCache, AdaptiveDiffusion, and SADA.

- Clarity and Theoretical Justification:

   - Reviewers noted that the writing quality is poor in areas, hindering understanding.

   - The theoretical justification for why residuals are inherently better prediction targets than features was found to be unclear.

   - Key definitions, such as the specific meaning of "Single," "Dual," and "Full" DiTBlocks, were ambiguous.

- Technical Constraints and Overhead:

   - Concerns regarding the memory consumption required to store residual histories were raised.

   - The paper lacks a detailed breakdown of latency overhead (computing curvature/gating vs. forward pass) to prove the speedups are not hardware-specific.

**Reviewer Scores:**

Given the absence of an author rebuttal, the reviewers did not have the opportunity to update their scores based on new information. Below is an assessment of how scores likely would have trended given the outstanding issues:

- Reviewer UBz3 (Current Score: 4):
   - Prediction: Maintain 4 or Drop to 3. This reviewer raised high-level concerns about the "manual/heuristic" nature of the decision signals and the missing distillation baselines. Without a rebuttal to justify the generalizability of the hyperparameters or provide the requested comparisons, the reviewer's skepticism regarding the method's robustness would likely solidify.


- Reviewer XAXC (Current Score: 2):

   - Prediction: Maintain 2. This reviewer identified fundamental issues with writing quality and the justification of the core premise (residuals vs. features). These are foundational issues that usually require a major revision to fix. Without a rebuttal clarifying the definitions (e.g., $\tilde{\kappa}_{t}$) and justifying the design choices, the score would arguably remain at a clear reject.

- Reviewer sQVm (Current Score: 6):
   - Prediction: Drop to 5 or 4. While this reviewer was initially positive about the empirical validation , they listed specific weaknesses regarding evaluation scope (scaling to 50/100 steps) and terminology confusion (Table 6). The lack of clarification on these points would likely lower their confidence in the paper's completeness and generalizability, moving them from a weak accept to a borderline/reject position.



- Reviewer 1vXq (Current Score: 4):
   - Prediction: Maintain 4. This reviewer requested a significant amount of additional experimental data, including solver sweeps and comparisons to four specific missing baselines. As these requests went unanswered, the reviewer would have no basis to improve their score and would likely maintain their assessment that the evaluation is insufficient.

---

### Decision · Program_Chairs · 2026-01-26

Reject